# The Trend of Scientific Productivity of Chinese, European Union, and United States Universities and Private Companies: Does the Future Belong to E-Technology Companies?

**Mauro G. Carta [1,\*], Matthias C. Angermeyer [2] and Silvano Tagliagambe [3,†]**

1   Department of Medical Sciences and Public Health University of Cagliari, 09042 Monserrato, Italy
2   Center for Public Mental Health, 3482 Gosing am Wagram, Austria; angermeyer@aon.at
3   Department of Architecture, Design and Urban Planning, University of Sassari, 07100 Sassari, Italy; sil.tagliagambe@gmail.com
\*   Correspondence: mgcarta@tiscali.it
†   Professor Emeritus.

**Abstract:** The purpose is to verify trends of scientific production from 2010 to 2020, considering the best universities of the United States, China, the European Union (EU), and private companies. The top 30 universities in 2020 in China, the EU, and the US and private companies were selected from the SCImago institutions ranking (SIR). The positions in 2020, 2015, and 2010 in SIR and three sub-indicators were analyzed by means of non-parametric statistics, taking into consideration the effect of time and group on rankings. American and European Union universities have lost positions to Chinese universities and even more to private companies, which have improved. In 2020, private companies have surpassed all other groups considering Innovation as a sub-indicator. The loss of leadership of European and partly American universities mainly concerns research linked to the production of patents. This can lead to future risks of monopoly that may elude public control and cause a possible loss of importance of research not linked to innovation.

**Keywords:** scientific productivity; universities; private companies; USA; European Union; China; innovation; research; societal

## 1. Introduction

A preliminary study [1] analyzed the trends of scientific productivity from 2015 to 2019 of the top 30 private companies and top universities according to the SCImago Institutions Ranking [2]. The aim was to develop a methodology for future research. However, the results suggested that academia might lose its role in competing with Internet-related private bodies, which are gaining relevance [1]. These suggestions were not in agreement with the results of a recent study [3] that showed private companies not directly interested in investing in research, but rather appeared to take advantage of university research work. Another study showed that patents held by private bodies rely on high-quality research carried out by academia [4]. The latter two studies analyzed data up to 2016, while the preliminary data of the paper by Carta and co-workers were from 2015 to 2019. Thus, it was hypothesized that the divergences may be due to a more recent trend owing to specifically emerging Internet-related companies. However, this hypothesis could have been advanced only if a longer period of time had been analyzed.

Furthermore, Carta's previous work took into consideration only European and American universities and did not include new emerging forces in academia. Given the particular strategic relevance of the topic, another interesting aspect would be the analysis of universities in relation to policy guidelines, therefore within state and supra-state bodies with homogeneous policies.

The purpose of this work is to verify trends in international research from 2010 to 2020, considering the best 30 universities of the United States of America, China, and the

European Union (the three major economic powers in the world) and to compare this trend with that of the best 30 private companies, in agreement with the SCImago 2020 ranking of scientific productivity [2].

Unlike the aforementioned preliminary research [1], this study will conduct a detailed analysis of all the sub-indicators considered by the SCImago ranking and will employ an adequate non-parametric analysis that will allow consideration of the effect of both time and group factors on rankings.

## 2. Materials and Methods

Design. To analyze the trends of scientific productivity from 2010 to 2020, the top 30 universities in 2020 in China, the European Union, and the United States were selected from the SCImago institutions ranking [2,5], as well as the top 30 private companies operating around the world. Then, the SIR positions of the same institutions in 2020, 2015, and 2010 and each of the three sub-indicators were analyzed and compared within each group; a comparison between groups was carried out for 2020, 2015, and 2010.

Indicators. The SCImago Institutions Rankings (SIR) is a "classification of academic and research-related institutions ranked by a composite indicator that combines three different sets of indicators based on research performance, innovation outputs and societal impact measured by their web visibility" [2] (SCImago 2020). Thus, SIR was the product of the following sub-indicators: (1) "Research", based on the number of articles published and the citations they attracted of a given institution (weighting of 50% of the total score) [2,6–11]; (2) "Innovation", consisting of the number of scientific publications of a given institution cited in applying for patents and the number of patent applications of the given institution in the PATSTAT databank [12] (weighting of 30% of the total score but a key dimension of the impact on the economy) [2,13]; (3) "Societal" calculated as the sums of (a) altimetrics (amount of documents with more than one mention in PlumX Metrics (https://plumanalytics.com, accessed on 10 August 2020) + amount of documents with more readers in Mendeley (https://www.mendeley.com, accessed on 15 July 2020), (b) amount of networks with links to the institution website, and (c) weight of the institution's URL according to Google [2,14]. The "Societal" indicator weighs 20% of the total. It was the last to be introduced, so it can be assessed only from 2015. It should also be noted that it has undergone some changes and is thus the least reliable for an analysis of temporal trends.

Statistical Analysis. We calculated the median position (±upper and lower quartile) by group (Chinese, European Union, and United States universities and private companies) for 2020, 2015, and 2010 in the SCImago Institutions Rankings and in the three sub-indicators. The position of the same institutions in 2020, 2015, and 2010 according to SIR, as well as each of the three sub-indicators, was compared by means of the Freidman test for repeated measures. The SIR comparison between groups in 2020, 2015, and 2010 and the three sub-indicators was carried out by means of the Kruskall–Wallis test. The use of non-parametric statistics on ordinal scale levels was due to the fact that we worked on ranks and not on variables measured as scales at equivalent intervals.

## 3. Results

Chinese, European Union, and United States universities and private companies occupying the top 30 steps in the SCimago ranking in 2020 are reported in Appendix A; of the European Union universities, 7 were from France and The Netherlands, 3 were from Germany, Sweden, Denmark and Italy, 2 were from Belgium, 1 was from Spain and Finland; among the private companies, 25 were from the US or dependents of US parent companies, 2 were from South Korea, 1 was from Sweden/UK, 1 was from Finland, and 1 was from France; of the private companies, 7 were pharmaceutical companies, 2 were biotechnology and pharmaceutical companies, 8 were Internet-related services; 8 focused on digital technologies, 4 focused on electronics and informatics with diversification, and

1 was a conglomerate of technologies, research, and finance. A total of 25 out of 30 private companies had pre-eminent interests in Internet and/or digital technologies.

Figure 1, Table 1 (difference between groups) and Table 2 (difference within groups) show the 2010–2020 trends on SCImago Institutions Rankings (SIR) of the four groups as found in 2020 in SCImago Rankings. Chinese universities increased their ranking over time, reaching a difference of statistical significance both from 2010 to 2015 (median 458 to 227, $p < 0.00001$) and 2015 to 2020 (median 227 to 171, $p < 0.00001$). Chinese universities were in third place and preceded private companies in 2010 (behind American and European universities). However, private companies grew more and Chinese universities therefore remained in last place in 2020 but reached European universities. European universities increased their rankings from 2010 to 2015 (median from 165.5 to 119.5, $p < 0.00001$) but decreased from 2015 to 2020 (median from 119.5 to 171.5 $p = 0.00120$). In conclusion, the rankings did not increase from 2010 to 2020. European universities were second behind the Americans in 2010, and they were in last place in 2020, together with Chinese universities (which however show a strong growth trend). American universities showed a progressive decrease from 2010 to 2015 (median from 32 to 45.5, $p < 0.00001$) and from 2015 to 2020 (median from 28 to 32, $p < 0.00001$). American universities were the leading group in 2010; this position has been maintained in 2020, but it is now shared with private companies. The position of private companies in the ranking has grown progressively, but with a wide distribution of the range, so although the median gradually decreases, the difference in the ranking is statistically significant from 2015 to 2020 (median from 69.5 to 218, $p < 0.00001$) but not from 2010 to 2015 (median from 715.5 to 218, $p < 0.067$). Private companies were in last place in 2010, but in 2020, they reached American universities in first place.

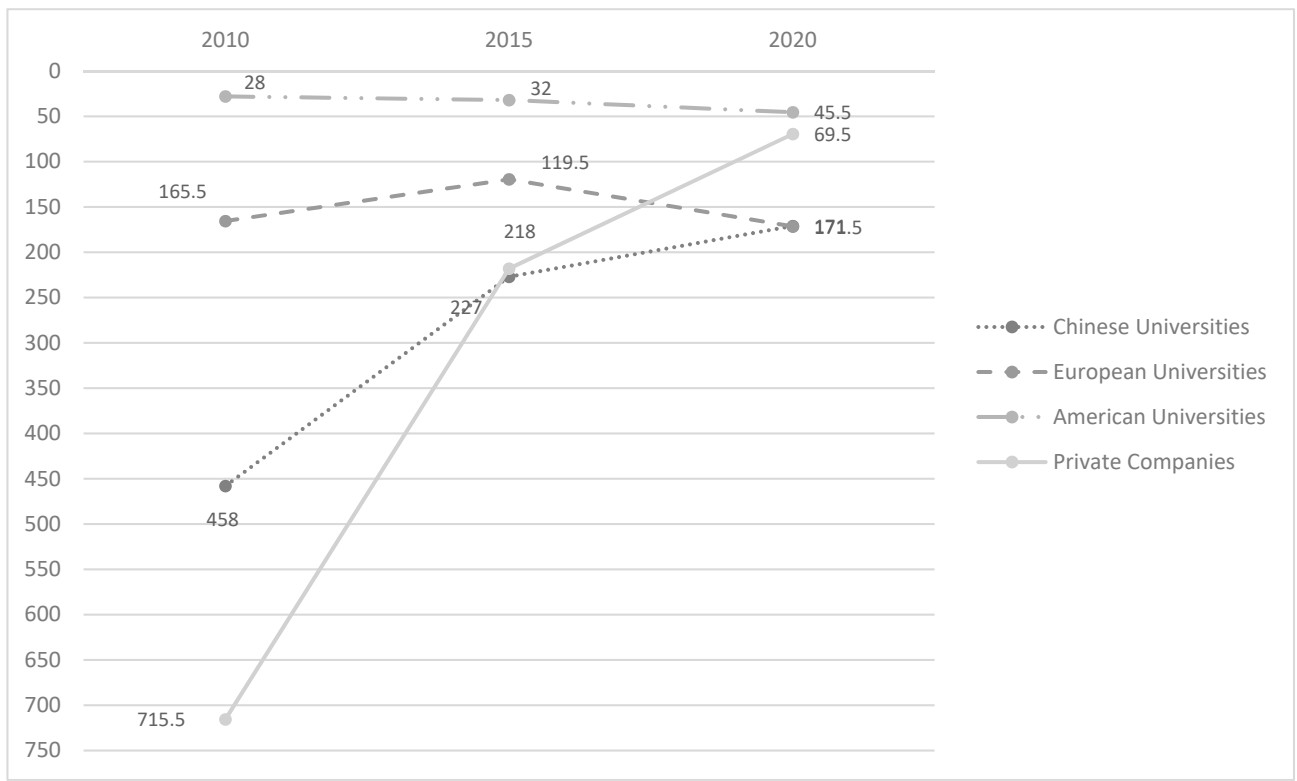

**Figure 1.** Trends 2010–2020 on SCImago institutions rankings (SIR) of the best 30 universities of China, the European Union, the US, and of the best 30 private companies.

Figure 2, Table 3 (difference between groups) and Table 4 (difference within groups) show the 2010–2020 trends in the same groups in the sub-indicator "Research". Chinese universities increased their ranking over time, reaching a difference of statistical significance both from 2010 to 2015 (median 320 to 130, $p < 0.00001$) and from 2015 to 2020

(median 130 to 104, $p = 0.00006$). Chinese universities from 3rd place in 2010 (behind American and European universities) reached European universities in 2nd place in 2020. European universities showed an increase in their ranking from 2010 to 2015 (median from 142.5 to 103.5, $p < 0.00001$). The trend reversed from 2015 to 2020 (median from 103.5 to 108.5, $p < 0.00001$) and, considering the whole 2010–2020 arc, the ranking has not changed. European universities were in 2nd place in 2010 (behind American universities), but in 2020, Chinese universities reached them. American universities showed a constant loss of positions in the ranking both from 2010 to 2015 (median from 33 to 35.5, $p = 0.0286$) and from 2015 to 2020 (median from 35.5 to 38, $p < 0.00001$), but they have maintained the former position over time. Private companies have recorded a constant decrease in their median ranking over time (722 in 2010, 304.5 in 2015, 291.5 in 2020) but with a wide range in distribution. Therefore, the improvement is not statistically significant, and private companies maintain the last ranking position over time.

**Table 1.** Trends 2010–2020 on SCImago institutions rankings (SIR) of the best 30 universities of China, the European Union, the US, and of the best 30 private companies.

| Group | 2020<br>Median ± Lower [$x_L$]/Upper [$x_U$] Quartile | 2015<br>Median ± Lower [$x_L$]/Upper [$x_U$] Quartile | 2010<br>Median ± Lower [$x_L$]/Upper [$x_U$] Quartile |
|---|---|---|---|
| Chinese Universities | 171, $x_L$ 106.25, $x_U$ 224 | 227, $x_L$ 133.75, $x_U$ 276.5 | 458, $x_L$ 269, $x_U$ 556.25 |
| European Universities | 171.5, $x_L$ 122.5, $x_U$ 207.5 | 119.5, $x_L$ 78.75, $x_U$ 141 | 165.5, $x_L$ 112.75, $x_U$ 223.25 |
| American Universities | 45.5, $x_L$ 30.5, $x_U$ 66.5 | 32, $x_L$ 17, $x_U$ 53.25 | 28, $x_L$ 14, $x_U$ 45 |
| Private Companies | 69.5, $x_L$ 27, $x_U$ 115.25 | 218, $x_L$ 93.25, $x_U$ 318.25 | 715.5, $x_L$ 97.5, $x_U$ 1000 |

Difference between groups in 2020, 2015, 2010 (Kruskal–Wallis Test) 2020: $H = 56.6201$ (3, $N = 120$). $p < 0.00001$. AU = PC ($H = 3.4966$ (1, $N = 60$) $p = 0.061$); AU + PC > CU + EU (CU = EU, $H = 0.0005$ (1, $N = 60$) $p = 0.982$); 2015: $H = 48.4697$ (3, $N = 119$), $p < 0.00001$. AU > EU + PC (EU = PC $H = 5.8074$ (1, $N = 60$), $p = 0.01596$); EU + $P$ > CU ($H = 10.3825$ (2, $N = 90$), $p = 0.0055$); 2010 $H = 51.5938$ (3, $N = 120$), $p < 0.00001$. AU > EU ($H = 29.923$ (1, $N = 60$) $p = 0.01596$), EU > CU + PC ($H = 8.3969$ (1, $N = 60$), $p = 0.0037$), CU = PC ($H = 1.744$ (1, $N = 60$), $p = 0.186$).

**Table 2.** Trends within groups over time (Friedman test for repeated measures).

| Group | 2020 vs. 2015<br>Friedman Test for r.m. | 2015 vs. 2010<br>Friedman Test for r.m. | 2020 vs. 2010<br>Friedman Test for r.m. |
|---|---|---|---|
| Chinese Universities | $X^2_r = 19.2$ (1, $N = 30$)<br>$p < 0.0001$ | $X^2_r = 26.133$ (1, $N = 30$)<br>$p < 0.0001$ | $X^2_r = 26.1333$ (1, $N = 30$)<br>$p < 0.0001$ |
| European Union Universities | $X^2_r = 10.8$ (1, $N = 30$)<br>$p = 0.00102$ | $X^2_r = 22.533$ (1, $N = 30$)<br>$p < 0.00001$ | $X^2_r = 0.5333$ (1, $N = 30$)<br>$p = 0.46521$ |
| American Universities | $X^2_r = 7.5$ (1, $N = 30$)<br>$p = 0.00617$ | $X^2_r = 3.333$ (1, $N = 30$)<br>$p = 0.06789$ | $X^2_r = 13.333$ (1, $N = 30$)<br>$p = 0.00026$ |
| Private Companies | $X^2_r = 22.533$ (1, $N = 30$)<br>$p < 0.00001$ | $X^2_r = 2.1333$ (1, $N = 30$)<br>$p = 0.14413$ | $X^2_r = 16.1333$ (1, $N = 30$)<br>$p = 0.00006$ |

**Table 3.** Trends 2010–2020 in sub-indicator "Research" in the best 30 universities of China, the European Union, the US, and in private companies.

| Group | 2020<br>Median ± Lower [$x_L$]/Upper [$x_U$] Quartile | 2015<br>Median ± Lower [$x_L$]/Upper [$x_U$] Quartile | 2010<br>Median ± Lower [$x_L$]/Upper [$x_U$] Quartile |
|---|---|---|---|
| Chinese Universities | 104, $x_L$ 69.25, $x_U$ 150 | 130, $x_L$ 86, $x_U$ 161.25 | 326, $x_L$ 209.25, $x_U$ 387.25 |
| European Universities | 108.5, $x_L$ 85.5, $x_U$ 128 | 103.5, $x_L$ 83.25, $x_U$ 134.25 | 142.5, $x_L$ 118.75, $x_U$ 190.25 |
| American Universities | 38, $x_L$ 20, $x_U$ 56 | 35.5, $x_L$ 18.25, $x_U$ 64.25 | 33, $x_L$ 15.25, $x_U$ 49.5 |
| Private Companies | 291.5, $x_L$ 153.5, $x_U$ 357.5 | 304.5, $x_L$ 165, $x_U$ 385 | 722, $x_L$ 173.75, $x_U$ 1000 |

Difference between groups (Kruskal-Wallis Test): 2020–$H = 49.8253$ (3, $N = 120$), $p < 0.00001$. AU > CU + EU [($H = 0.0369$ (1, $N = 60$) $p = 0.84759$]. PC + EU > PC [$H = 31.0483$ (2, $N = 909$, $p < 0.00001$], 2015–$H = 61.3979$ (3, $N = 120$), $p < 0.00001$. AU > EU + CU > PC [EU = CU ($H = 3.2002$ (1, $N = 60$), $p = 0.0736$], 2010–$H = 52.2706$ (3, $N = 120$), $p < 0.00001$. AU > EU [$H = 26.4708$ (1, $N = 60$), $p < 0.00001$]; EU > CU [$H = 7.8079$ (1, $N = 60$), $p = 0.0052$]; CU > PC [$H = 4.7233$ (1, $N = 60$), $p = 0.02976$].

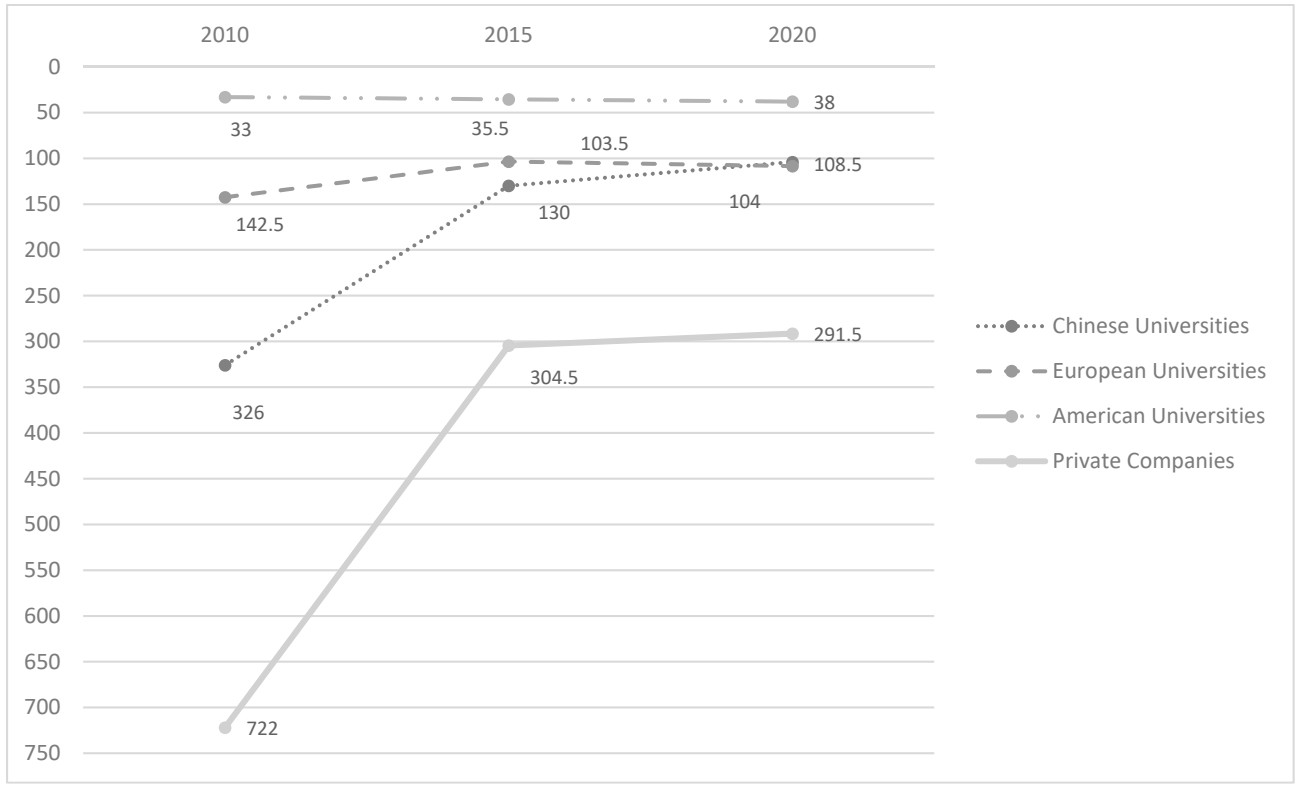

**Figure 2.** Trends 2010–2020 in sub-indicator "Research" in the best 30 universities of China, the European Union, the US, and in private companies.

**Table 4.** Trends within groups over time (Friedman test for repeated measures) in sub-indicator "Research" in the best 30 universities of China, the European Union, the US, and in private companies.

| Group | 2020 vs. 2015 Friedman Test for r.m. | 2015 vs. 2010 Friedman Test for r.m. | 2020 vs. 2010 Friedman Test for r.m. |
|---|---|---|---|
| Chinese Universities | $X^2 = 16.1333$ (1, $N = 30$) $p < 0.00006$ | $X^2 = 30$ (1, $N = 30$) $p < 0.00001$ | $X^2 = 26.1333$ (1, $N = 30$) $p < 0.00001$ |
| European Union Universities | $X^2_r = 26.133$ (1, $N = 30$) $p < 0.00001$ | $X^2_r = 20.833$ (1, $N = 30$) $p < 0.00001$ | $X^2_r = 0$ (1, $N = 30$) $p = 1$ |
| American Universities | $X^2_r = 17.633$ (1, $N = 30$) $p = 0.00003$ | $X^2_r = 4.8$ (1, $N = 30$) $p = 0.02846$ | $X^2_r = 19.2$ (1, $N = 30$) $p < 0.00001$ |
| Private Companies | $X^2_r = 1.2$ (1, $N = 30$) $p = 0.27332$ | $X^2_r = 0.1333$ (1, $N = 30$) $p = 0.715$ | $X^2_r = 1.2$ (1, $N = 30$) $p = 0.27332$ |

Figure 3, Table 5 (difference between groups) and Table 6 (difference within groups) show the 2010–2020 trends for the sub indicator "Innovation". Chinese universities showed an increase in ranking from 2010 to 2015 (median from 395.5 to 291.5, $p < 0.00001$). This trend finished from 2015 to 2020 (median from 291.5 to 287, difference not statistically significant), but considering the whole 2010–2020 arc, the ranking improved ($p < 0.00001$). Chinese universities were in last place in 2010, but overtook European universities in 2020, while private companies improved even more.

European universities increased their rankings from 2010 to 2015 (median from 164 to 116, $p < 0.00001$) but decreased strongly from 2015 to 2020 (median from 116 to 324.5, $p < 0.00001$). In conclusion, the rankings did not increase from 2010 to 2020. European universities were second in 2010 behind American ones, but in 2020, Chinese universities and private companies surpassed them. American universities showed a trend of loss of

positions between 2010 and 2015, but the difference between the two rankings did not reach statistical significance (median from 28 to 33.5, *p* = 0.715). This trend accentuated from 2015 to 2020, reaching a clear worsening trend (median from 33.5 to 153.5, *p* = 0.00026). The difference between the 2010 and 2020 rankings is also statistically significant, thus indicating a general trend toward worsening (*p* = 0.00102). The American universities, that were in first place in 2010 have been surpassed by private companies in 2020. These companies showed a huge improvement between 2010 and 2015 (median from 605.5 to 143, *p* < 0.00001), which continued between 2015 and 2020 (median from 143 to 28, *p* = 0.002846). Private companies moved from last place among the four groups considered in 2010 to the first in 2020.

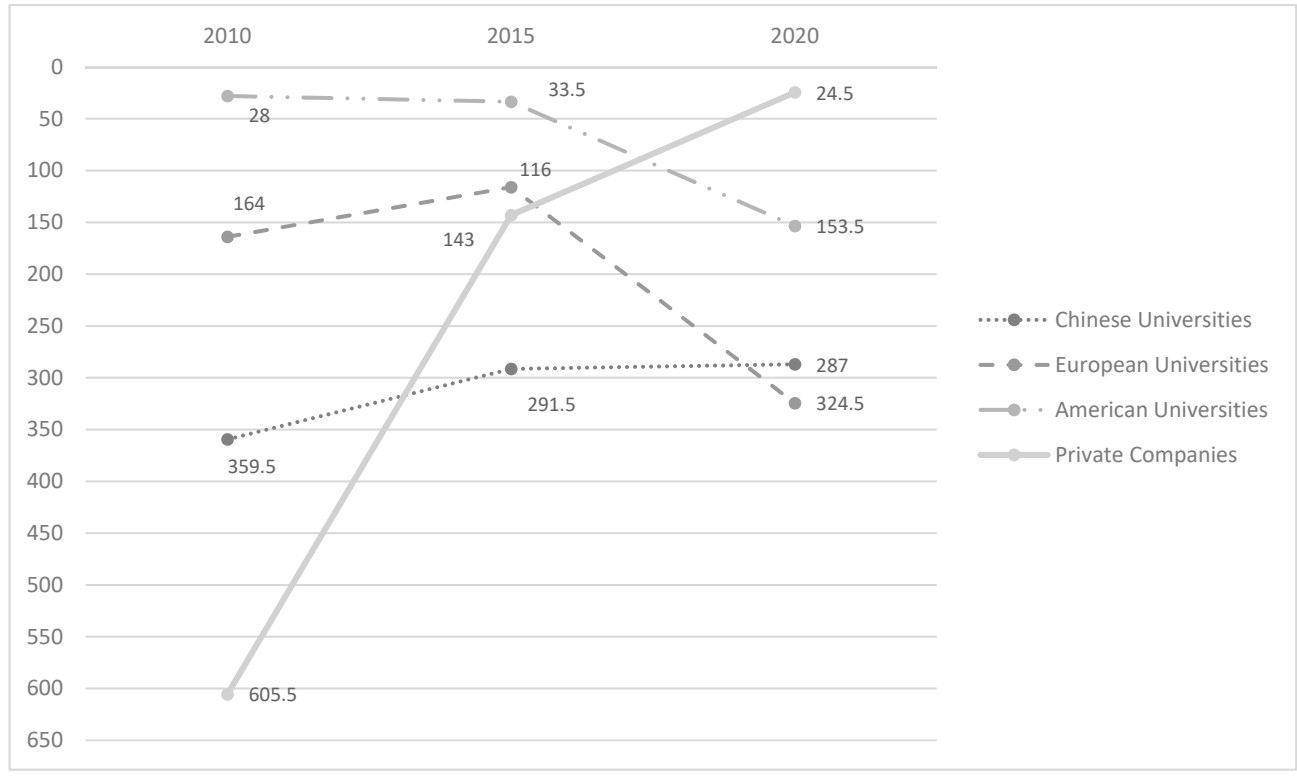

**Figure 3.** Trends 2010–2020 in sub indicator "Innovation" in the best 30 universities of China, the European Union, the US, and in private companies.

**Table 5.** Trends 2010–2020 between groups in sub indicator "Innovation" in the best 30 universities of China, the European Union, the US, and in private companies.

| Group | 2020 Median ± Lower [$x_L$]/Upper [$x_U$] Quartile | 2015 Median ± Lower [$x_L$]/Upper [$x_U$] Quartile | 2010 Median ± Lower [$x_L$]/Upper [$x_U$] Quartile |
|---|---|---|---|
| Chinese Universities | 287, $x_L$ 233.25, $x_U$ 314 | 291.5, $x_L$ 186.5, $x_U$ 320 | 359.5, $x_L$ 226.75, $x_U$ 393.75 |
| European Universities | 324.5, $x_L$ 298.75, $x_U$ 355.25 | 116, $x_L$ 78.75, $x_U$ 201.25 | 164, $x_L$ 101, $x_U$ 210 |
| American Universities | 153.5, $x_L$ 22.5, $x_U$ 221.25 | 33.5, $x_L$ 13.75, $x_U$ 54.25 | 28, $x_L$ 15.25, $x_U$ 53.75 |
| Private Companies | 24.5, $x_L$ 10.25, $x_U$ 42.5 | 143, $x_L$ 37, $x_U$ 282.5 | 605.5, $x_L$ 87.5, $x_U$ 1000 |

Difference between groups (Kruskal–Wallis Test). 2020: *H* = 84.2197 (3, *N* = 120), *p* < 0.00001; PC > AU, (*H* = 15.2919 (1, *N* = 60), *p* < 0.00001); AU > CU (*H*) 40.6035 (1, *N* = 60), *p* < 0.00001); CU > EU (*H* = 10.9674 (1, *N* = 60), *p* = 0.00093); 2015: *H* = 57.7482 (3, *N* = 120). *p* < 0.00001. AU > EU + PC (*H* = 30.4965 (2, *N* = 90), *p* < 0.00001); EU = PC (*H* = 0.0831 (1, *N* = 60), *p* = 0.77312); EU + PC > CU (*H* = 19.9271 (2, *N* = 90), *p* < 0.00005; 2010: *H* = 48.8813 (3, *N* = 120), *p* < 0.00001. AU > EU (*H* = 29.463 (1, *N* = 60), *p* < 0.00001); EU > CU + PC (*H* = 6.7959 (2, *N* = 90); *p* = 0.03344); CU = PC (*H* = 0.0874 (1, *N* = 60), *p* = 0.76747).

**Table 6.** Trends within groups over time (Friedman test for repeated measures) in sub indicator "Innovation" in the best 30 universities of China, the European Union, the US, and in private companies.

| Group | 2020 vs. 2015 Friedman Test for r.m. | 2015 vs. 2010 Friedman Test for r.m. | 2020 vs. 2010 Friedman Test for r.m. |
|---|---|---|---|
| Chinese Universities | $X^2 = 0$ (1, $N = 30$) $p = 1$ | $X^2 = 19.2$ (1, $N = 30$) $p < 0.00001$ | $X^2 = 17.6333$ (1, $N = 30$) $p < 0.00003$ |
| European Union Universities | $X^2_r = 26.133$ (1, $N = 30$) $p < 0.00001$ | $X^2_r = 20.833$ (1, $N = 30$) $p < 0.00001$ | $X^2_r = 0$ (1, $N = 30$) $p = 1$ |
| American Universities | $X^2_r = 13.333$ (1, $N = 30$) $p = 0.00026$ | $X^2_r = 0.1333$ (1, $N = 30$) $p = 0.715$ | $X^2_r = 10.8$ (1, $N = 30$) $p = 0.00102$ |
| Private Companies | $X^2_r = 26.1333$ (1, $N = 30$) $p < 0.00001$ | $X^2_r = 4.8$ (1, $N = 30$) $p = 0.02846$ | $X^2_r = 22.5333$ (1, $N = 30$) $p < 0.00001$ |

Figure 4 and Tables 7 and 8 show the 2015–2020 trends in the same groups for the sub-indicator "Societal", which was not collected in 2010. In this indicator, all groups considered showed a decrease in their ranking (median CU from 27 to 138.5, $p < 0.00001$; median EU, from 27 to 98.5, $p < 0.00001$; median AU from 25 to 40.5, $p < 0.00001$) with the exception of private companies in which the difference in the distribution of the rankings in the two surveys did not reach statistical significance, despite a sharp increase in the median (from 28 to 126). However, it is to be noted (see Tables 7 and 8) that the lower quartile of the distribution remains unchanged, while the median worsens slightly. The comparison between groups shows no substantial differences in the ranking over time, with the American universities in first place both in 2015 and 2020 and the private companies last in both evaluations, but with a first quartile with excellent performance in 2020 (almost similar to American universities).

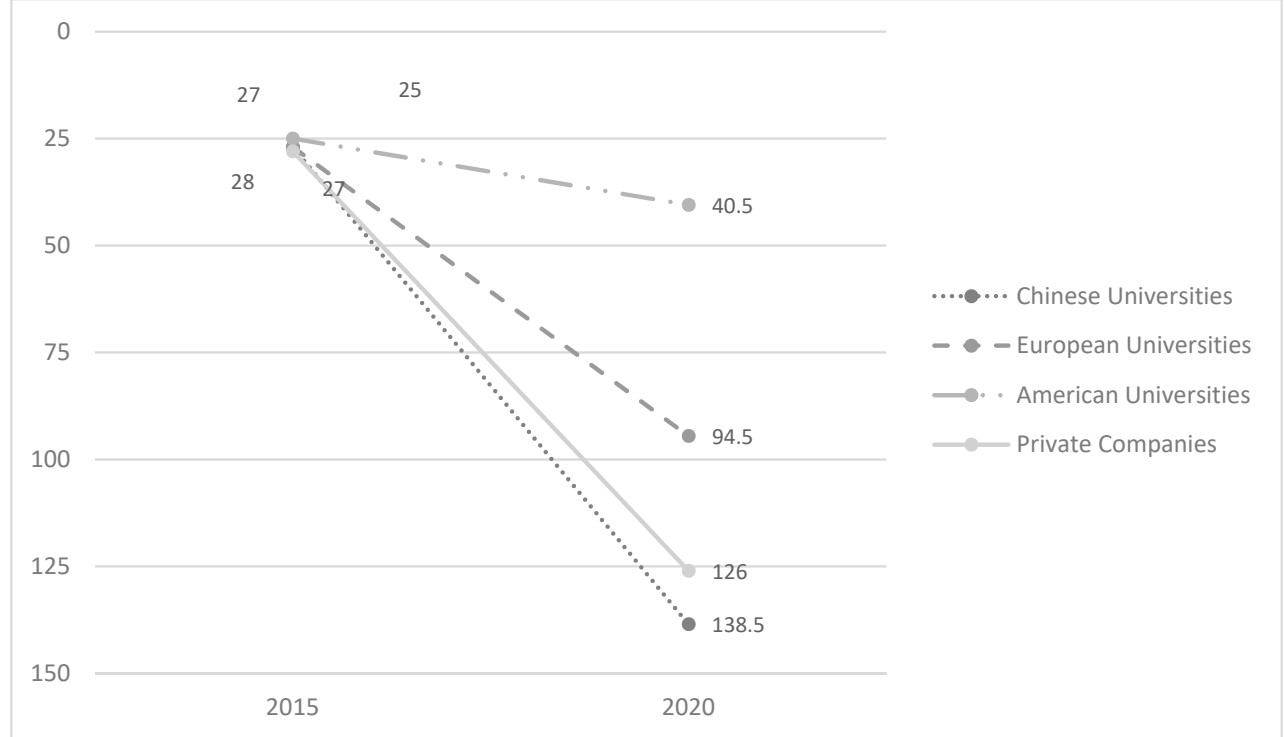

**Figure 4.** Trends 2010–2020 in sub-indicator "Sociality" in the best 30 universities of China, the European Union, the US, and in private companies.

**Table 7.** Trends between groups 2010–2020 in sub-indicator "Sociality" in the best 30 universities of China, the European Union, the US, and in private companies.

| Group | 2020 Median ± Lower [$x_L$]/Upper [$x_U$] Quartile | 2015 Median ± Lower [$x_L$]/Upper [$x_U$] Quartile |
|---|---|---|
| Chinese Universities | 138.5, $x_L$108, $x_U$ 160.5 | 27, $x_L$ 27, $x_U$ 28 |
| European Universities | 94.5, $x_L$ 81.25, $x_U$ 102.25 | 27, $x_L$ 24, $x_U$ 28 |
| American Universities | 40.5, $x_L$ 26.75, $x_U$ 63.50 | 25, $x_L$ 20, $x_U$ 27.25 |
| Private Companies | 126, $x_L$ 28, $x_U$ 221 | 28, $x_L$ 28, $x_U$ 177.25 |

Difference between groups (Kruskal–Wallis Test). 2020: $H = 52.5374$ (3, $N = 120$), $p < 0.0001$; AU > EU + PC [$H = 11.3056$ (2, $N = 90$); $p = 0.00351$]; EU > PC [$H = 5.2514$; $p = 0.02193$]; PC > CU [$H = 4.1325$ (1, $N = 60$), $p = 0.04207$]. 2015: $H = 35.0568$ (3, $N = 120$), $p < 0.00001$. AU > CU + EU [$H = 13.5851$ (2, $N = 90$), $p = 0.00112$]; CU = EU [$H = 2.7664$ (1, $N = 60$), $p = 0.09626$]: CU + EU > PC [$H = 20.7775$ (2, $N = 90$), $p = 0.00003$].

**Table 8.** Trends within groups over time in sub-indicator "Sociality" in the best 30 universities of China, the European Union, the US, and in private companies Friedman test for repeated measures).

| | 2020 vs. 2015 Friedman Test for r.m. |
|---|---|
| Chinese Universities | $X^2 = 30$ (1, $N = 30$) $p < 0.0001$ |
| European Union Universities | $X^2_r = 19.2$ (1, $N = 30$) $p < 0.0001$ |
| American Universities | $X^2_r = 16.1333$ (1, $N = 30$) $p < 0.0001$ |
| Private Companies | $X^2_r = 0.3$ (1, $N = 30$) $p = 0.58388$ |

## 4. Discussion

This study has found that American and European Union universities are losing positions in rankings of scientific productivity as measured by the SCimago website. This trend was not observed for Chinese universities and even more so for private companies which, on the contrary, improved their performance. The loss of scientific productivity in European universities was detected from 2015 to 2020, while in previous periods, an improvement was still noted. Instead, the loss of scientific productivity in American universities is stable in the two surveys (2010–2015 and 2015–2010), although it is less marked. The improvement in scientific productivity in private companies appeared from 2015 to 2020. Instead, the improvement in scientific productivity in Chinese universities is stable in the two surveys, although it is less marked. A similar but less important trend can be seen in the "Research" sub-indicator, where the trend of increasing productivity in private companies over time does not reach statistical significance, and these remain in last place of the four groups analyzed in all three assessments over time. In fact, this indicator includes all scientific productivity and therefore also sectors very far from innovation and commercial interest: not only basic research but also all those fields, from history to literature, which do not directly involve, or which involve indirectly patents and elements that influence the market.

Universities maintain prominence in this sub-indicator, at least in part as to why in certain sectors basic research (even that which is essential for translational research) is still delegated to the universities. This has recently been confirmed by research on Covid-19 vaccines [15,16]. Therefore, these results seem not totally in contrast with the previously cited study, which had suggested a complementary role of public and private research [3], even if the importance of the universities still seems to be downsized compared to the past.

A quite different result emerges from the analysis of the Innovation indicator: here, private companies have a constant and almost exponential growth and surpass all other groups in 2020. The American and European universities that increased up to 2015 suffered a weakening of their positions only after that period. Chinese universities have shown a steady but moderate growth.

Therefore, the data of our study confirm an inversion of the trend of private companies which in past years were not interested in investing directly in research but rather in exploiting the work of universities to obtain patents, as evidenced by studies that analyzed the production of patents until 2016 [3,4]. However, this trend is compatible with our results, which show a progressive increase since 2010 in the direct action of private companies in the production of patents. In fact, in the four groups considered, private companies are in last place, with Chinese universities in 2010 in the sub-indicator "Innovation". They reached European universities in 2015 with equal merit in second place; they surpassed all universities, including the Americans, in 2020. Therefore, the reversal of the trend has become evident only in recent years. It is to be noted that the protagonists of this progressive increase in the direct production of patents were 14 out of 30 companies which in 2010 were not even present among the first 1000 institutions in the ranking and that in 2020 are now among the first 30. These 14 are all companies that produce Internet services or digital technologies. Among these were included five of the seven companies that provide Internet services, which in 2020 were among the top 10 in the ranking. Therefore, the study appears to support the hypothesis that the change in trend is mainly due to the entry into the research scene of companies linked to the Internet and the production of digital technologies.

Our results point out that the increased importance of the scientific production of private companies is mainly due to applied and translational research. However, this sector is increasingly "weighing" on general scientific productivity [5]. This may imply a loss of importance of basic research and related skills and knowledge that in the long term may have negative implications on innovation itself [17].

The loss of predominance of universities puts the idea (or the myth) of nineteenth-century Western society about research and the role of universities in a serious crisis. In fact, this vision saw science (and universities as a place dedicated to the development of science) as an expression of free thought, which is useful in furthering human development from which everyone can benefit [18] (Von Humboldt 1810).

Our data show that private companies have been gaining increasing importance in the research field with Internet-related services and digital technology companies as the protagonists. Notably, in the recent past, the top ranking companies have been accused by institutional voices of the EU and the US of using data gathered indiscriminately and amorally for their own benefit and social control [19,20]. All this recalls the pessimism of the Dialectic of Enlightenment [21]: science that was believed to support emancipation and freedom may today become a tool furthering the interests of single companies with little chance of social control. However, this result clearly highlights that European universities are currently the weakest. These institutions have lost considerable visibility in the last 5 years, and the negative balance reflects even more the strategic indicator "Innovation".

The "Societal" indicator appears to be influenced by the fact that it is not yet consolidated because it was introduced very recently. This makes it difficult to analyze it in depth. First of all, this sub-indicator has undergone recent improvements and modifications. Moreover, the SCImago rankings do not include the progressive numbers that share the same ranking in the calculation: for example, if 10 institutions share ranking number 1, the next ranking is not the number 11 but 2. This has little influence if the calculation is very complex and there are few draws, but as regards the Sociality sub-indicator, in the first evaluation after it was introduced, the measures that produced the indicator were few (it was only in the following years that the indicator was better defined), so there were many draws (in fact in 2015 the ranking was quite low). The indicator has made changes, and therefore, evaluation over time is problematic.

One relevant issue that our study brings to attention is the dramatic loss of positions of EU universities. It may be the result of a more general crisis concerning the role of research in Europe. Several elements underscore a critical moment for research in Europe and perhaps that must be addressed. In 2015, resources dedicated to research were 2.0% of the overall GDP in Europe in contrast to 2.7% of the US and 2.1% of China, but in China, there was a 60% increase in funding from 2000 to 2015, while in Europe, the growth was 15% [22,23]. Further problems will likely intensify the scarcity of funds.

In the United States, investments in research have a direct impact on the production of highly cited scientific articles and the number of patents [24].

This relationship does not seem to be as linear in Europe [25]; in fact, a study compared the scientific impact of two matched groups of funded projects in the field of active aging, one from EU-WP7 and the other from US-NIH, and it found that the same number of scientific publications was produced for the two groups of projects, which showed a similar amount of citations [26]; however, the analyzed sample of EU-WP7 projects costs ten times more than the US-NIH ones [27,28].

In addition to general problems inherent in the weight of research, there are other issues in Europe (or at least in many countries of the current European Union) that are specific to the university and amplify the general weakness concerning research in the specific framework of academia. First of all, most universities in Europe are public and in Europe, especially in southern Europe, "the limitation of public funding following the economic crisis in 2008 has put greater pressure on their public universities to achieve excellence and improve competitiveness" [29]. This concerns funds dedicated to the maintenance of university facilities, not the research funds that we mentioned previously. This dramatic decrease in investments has been amplified by the protectionism prevailing in Italy, which causes promising young researchers to take flight, resulting in brain drain [30,31]. However, even in countries where the crisis has been more easily overcome, structural issues of universities are spotted as the hierarchical structure of German and Austrian universities [32–34]. Another problem is low intergenerational mobility, which prevents better exploitation of the pool of talents [35]. Some reports have also underscored a difficulty in changing the organizational structure of tertiary education in post-communist European Union countries [36]. This is probably why in the first 30 European universities for scientific productivity, no one is in a post-communist country.

## 5. Limits

At the present time, SCImago is the only database that offers scientific productivity outcomes grouped by institutions. Therefore, the use of other sources would be much more cumbersome and problematic. This represents a limitation, since in fact SCImago is not exempt from criticism [37,38]; it has been written that SCimago "omits a large amount of information, putting into question its transparency, reliability, and suitability for evaluative purposes in its current form, although most of the identified problems can be solved and might be the object of future improvements" [39,40].

In addition to the limitations of SCimago, there are also the intrinsic limits to these bibliometric methodologies, which lie in the assumption that the hundreds of thousands of publications with multiple authors can be neatly attributed to hundreds of institutions and that the role of self-citations can somehow distort the results. These are not the failings of the present study but the limits of this approach with the current tools.

We also need to underline the way in which we have ranked European universities in relation to Brexit (and the UK Universities). The survey was conducted with the aim of understanding the trend in scientific productivity of the academia of the three major nation or federations of the world (by economic and scientific power) and of private companies. At the present time and for future projections of the European Union, this federation has to do without UK universities (from 1 February 2020, date of Brexit). Of course, if we consider the universities of the United Kingdom in 2010 and 2015, the position of the European Union would be higher than without the UK (we have considered it without the UK) and

therefore, the decrease would be even greater. In fact, a clear decrease also emerges in our analysis that did not consider UK universities even in the past. Therefore, our study underlines that the tendency to lose positions in the European Union's world productivity ranking is also independent of Brexit. Furthermore, a multivariate analysis (taking into account the trend with and without the UK) would be very problematic having to work on ranks and not on absolute values. Future studies would be very helpful in clarifying just how much, and if Brexit has may have weakened research in the European Union (and the United Kingdom).

The purpose of this work was to draw an overall picture, and for this reason, we could not go into detail in the analysis of those institutions that appear to be in contrast with the general trend, as well as those in which the trend appears to be very accentuated. The role of joint ventures with private companies in the first cases mentioned and the greater impact of the decrease in public funding in geographic poor areas in the second deserve to be analyzed first with specific case reports and then with research that adopts a more specifically prepared methodology in the future.

## 6. Conclusions

The study describes a loss of leadership from 2010 to 2020 of European and partly American universities, while in China, universities are growing in importance in scientific production, and we are witnessing a growth of direct leadership in the research of private companies. This phenomenon mainly concerns research linked to the production of patents. It may lead to future risks of monopoly with difficulties for the public sector to lead and finalize research and to a possible loss of importance of research not linked to innovation.

**Author Contributions:** The study was initially designed by M.G.C. and then discussed with S.T. and M.C.A. The methodology was decided by M.G.C., S.T. and M.C.A., M.G.C. conducted the data analysis. The results were discussed collectively. M.G.C., S.T. and M.C.A. drafted the paper. All authors have read and agreed to the published version of the manuscript.

**Funding:** The study did not receive external funding.

**Institutional Review Board Statement:** Not applicable.

**Informed Consent Statement:** Not applicable.

**Data Availability Statement:** The datasets used for the analyses are available at the following links: SCImago Institutions Ranking 2020 (https://www.scimagoir.com/rankings.php, accessed on 5 June 2020); SCImago Institutions Ranking 2015 (https://www.scimagoir.com/rankings.php?year=2009, accessed on 5 June 2020); SCImago Institutions Ranking 2010 (https://www.scimagoir.com/rankings.php?year=2004, accessed on 5 June 2020).

**Conflicts of Interest:** The authors declare that they have no competing interests.

## Appendix A

**Table A1.** Chinese, European Union, and United States universities and private companies occupying the top 30 spots in the SCimago ranking.

| | Chinese Universities | | European Union Universities | | US Universities | | Private Companies |
|---|---|---|---|---|---|---|---|
| 1. | Tsinghua University. | 1. | Sorbonne Université (Fr). | 1. | Harvard University. | 1. | Facebook US. |
| 2. | Pekin University. | 2. | Université de Paris (Fr). | 2. | Harvard Medical School. | 2. | Facebook Inc. |
| 3. | Hong Kong University. | 3. | Catholic University of Leuven (Bel). | 3. | Massachusetts Institute of Technology. | 3. | Google Inc., USA. |
| 4. | Shanghai Jiao Tong University. | 4. | University of Copenhagen (DK). | 4. | Stanford University. | 4. | Microsoft, USA. |

**Table A1.** *Cont.*

| | Chinese Universities | | European Union Universities | | US Universities | | Private Companies |
|---|---|---|---|---|---|---|---|
| 5. | Zhejiang University. | 5. | Utrecht University (NLD). | 5. | Johns Hopkins University. | 5. | Microsoft Corp. |
| 6. | University of Chinese Academy of Sciences. | 6. | University of Amsterdam (NLD). | 6. | University of Washington. | 6. | Samsung Corp. |
| 7. | Huazhong University of Science and Technology. | 7. | Karolinska Institute (Swe). | 7. | University of Michigan, Ann Arbor. | 7. | Google International LLC. |
| 8. | Fudan University. | 8. | Technische Universitat Munchen (Ger). | 8. | Howard Hughes Medical Institute. | 8. | DeepMind Technologies GB. |
| 9. | Harbin Institute of Technology. | 9. | Ghent University (Bel). | 9. | University of Pennsylvania. | 9. | Microsoft Res, Asia, CHN. |
| 10. | Tianjin University. | 10. | Ecole Pratique des Hautes Etudes (Fr). | 10. | University of California, Los Angeles. | 10. | Samsung Electronics, SKor. |
| 11. | Sichuan University. | 11. | University of Groningen (NDL). | 11. | University of California, San Diego. | 11. | IBM USA. |
| 12. | Sun Yat-Sen University. | 12. | VU University Amsterdam (NDL). | 12. | Broad Institute of MIT and Harvard. | 12. | IBM Research. |
| 13. | ilin University. | 13. | Università degli Studi di Roma La Sapienza (Ita). | 13. | Columbia University. | 13. | Alphabet Inc. |
| 14. | University of Science and Technology of China. | 14. | Centre Roland Mousnier (FRA). | 14. | University of California, Berkeley. | 14. | Regeneron Pharmaceuticals Inc. USA. |
| 15. | Xi'an Jiaotong University. | 15. | Leiden University (NDL). | 15. | Cornell Univesity. | 15. | Hoffmann-La Roche, Ltd., US. |
| 16. | Nanjing University. | 16. | Aarhus University (DNK). | 16. | Whitehead Institute for Biomedical Research. | 16. | F Hoffmann-La Roche. |
| 17. | Shandong University. | 17. | Ludwig-MaximiliansUniversitat Munchen (Ger). | 17. | University of California, San Francisco. | 17. | Hoffmann-La Roche, Germany. |
| 18. | Central South University. | 18. | Erasmus University Rotterdam (NDL). | 18. | Massachusetts General Hospital. | 18. | Genentech Inc. US. |
| 19. | South China University of Technology. | 19. | Lunds University (SWE). | 19. | Yale University. | 19. | Microsoft Research, UK. |
| 20. | Wuhan University. | 20. | Uppsala University (Swe). | 20. | Duke University. | 20. | Qualcomm Inc., US. |
| 21. | Southeast University, Nanjing. | 21. | Technical University of Denmark (DNK). | 21. | University of Minnesota, Twin Cities. | 21. | Qualcomm Inc. |
| 22. | Tongji University. | 22. | University of Helsinki (FIN). | 22. | Brigham and Women's Hospital. | 22. | MedImmune, LLC. US. |
| 23. | Soochow University, Suzhou. | 23. | Aix-Marseille Université (Fr). | 23. | Ragon Institute of MGH, MIT and Harvard. | 23. | Novartis Institutes for Biomedical Research. US. |

**Table A1.** *Cont.*

| Chinese Universities | | European Union Universities | | US Universities | | Private Companies | |
|---|---|---|---|---|---|---|---|
| 24. | Nankai University. | 24. | Universita degli Studi di Milano (Ita). | 24. | University of Wisconsin, Madison. | 24. | Nokia Corp. |
| 25. | BeiHang University. | 25. | Universitat Heidelberg (Ger). | 25. | Northwestern University, Evanston. | 25. | Intel Corp., US. |
| 26. | The Chinese University of Hong Kong. | 26. | Université Paris-Saclay (Fr). | 26. | Scripps Research Institute. | 26. | Intel Corp. Inter. |
| 27. | University of Electronic Science and Technology of China. | 27. | Universita degli Studi di Padova (Ita). | 27. | Harvard-MIT Division of Health Sciences and Technology. | 27. | NVIDIA Corp. US. |
| 28. | Dalian University of Technology. | 28. | Universitat de Barcelona (ESP). | 28. | University of North Carolina, Chapel Hill. | 28. | Gilead Sciences Inc., US. |
| 29. | Xiamen University. | 29. | Universite Grenoble-Alpes (Fr). | 29. | Mayo Clinic. | 29. | Alcatel-Lucent. |
| 30. | Chinese Academy of Medical Sciences and Peking Union Medical College. | 30. | Delft University of Technolog (NDL). | 30. | University of Maryland, Baltimore. | 30. | Gilead Sciences. |

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
