# Peer review of "The Trend of Scientific Productivity of Chinese, European Union, and United States Universities and Private Companies: Does the Future Belong to E-Technology Companies?"

_publications, doi:10.3390/publications9020018_

Round 1
Reviewer 1 Report
The issue of academic productivity is one of the determinants of the scientific maturity of both the individuals conducting research and the institutions they represent. Productivity, i.e. the number of published texts indexed in significant periodicals, is an indicator of a researcher's status. Scientists are increasingly racing to publish their research results in prestigious journals. The pursuit of visibility in top journals (preferably q1) has its advantages and disadvantages. The authors of this article have focused on the foundational data offered by SCImago. This is a data repository affiliated with the Elsevier consortium. Currently SCImago, next to Claritive (Web of Science), it is one of the two organisations enjoying the greatest prestige among researchers from all over the world. The authors have shown very important trends. The article should be recommended for publication.
Of course, the text may be challenged and debatable due to the narrowness of the factors analysed. Nevertheless, the authors have presented the trends very clearly, which can be a valuable reference for further discussion. I rarely recommend a text in the first round for acceptance. This time I do so with complete conviction.
Author Response
We thank the reviewer for the comments. In fact, he does not ask us for changes.
However, since the reviewer says "Of course, the text may be challenged and debatable due to the narrowness of the factors analyzed", we reply that, also in relation to the suggestions of another reviewer, we have expanded the part dedicated to the limits.
The English has been revised
Reviewer 2 Report
The authors explore a rather timely issue, namely scientific productivity by comparing the United States, China, EU universities and top companies worldwide in the past decade. The main aspect of this comparison between academia and the business sphere is the quantification of scholarly achievement according to such indicators as the number of articles and the respective citations, complemented with the contribution to innovation, and the web-based presence and visibility of the given institutions The findings are summarized in several tables which are accompanied with thorough statistical analysis. The authors justify their theme selection, namely the performance of the given higher education institutions and business enterprises, and provide excessive data to substantiate their choices. The statistical analysis has three main reference points, 2010, 2015, and 2020. The authors deploy a variety of statistical approaches (Kruskal-Wallis test, Friedman test) and in addition to looking at the three abovementioned time periods they perform comparative analyses as well. The article’s conclusion implies the decreasing productivity and international weight of universities in the EU and to a limited extent in the United States compared to that of Chinese higher educational institutions and private enterprises.
Gray text highlights in the text are confusing and should be removed. The abstract is a literal version of an earlier work from https://www.preprints.org/subject/browse/social_sciences/organizational_economics_manag, the entire abstract needs to be rewritten.

Author Response
Gray texts have been removed.
The only other change required is the rewriting of the abstract because it has already been published.
However, the reviewer did not realize that the previously published abstract is that of the pre-print version of this same publication. That is, when we submitted the paper to this journal, the publisher's site itself asked us if we agreed to publish it as pre-print. We accepted so the site automatically produced that publication cited by the reviewer. So, I imagine it's not plagiarism as it is officially the pre-print of this article. The pre-print was produced as part of the submission and therefore with the consent of the publisher.
Reviewer 3 Report
All attempts to reduce the chaos of organizations to unified metrics are open to critique. There is a fairly extensive critical literature on the strengths and weaknesses of SCImago although it has not been given adequate attention here.
Obvious problems that come without comment here are 1. the definition of Europe; it makes no sense that there is not a single university from the UK in the analysis [it may have left the EU but was there when the initial rankings were undertaken]; 2. the inclusion of medical corporations such as Mayo in the university column, when it should be in the corporate column; 3. the assumption that the hundreds of thousands of publications with multiple authors can be neatly attributed to hundreds of institutions. These are not the failings of the authors but they do indicate the limits to this taxonomy.
All in all the results seem uninteresting and predictable as they stay at the level of the 4 groups [China, EU, US, Corporate]. To introduce some novelty, the authors could get into the weeds and for instance compare and contrast a single organization from each column and assess their contributions in terms of research and social impact.
Author Response
All attempts to reduce the chaos of organizations to unified metrics are open to critique. There is a fairly extensive critical literature on the strengths and weaknesses of SCImago although it has not been given adequate attention here.
Answer
We have introduced a section on limits reporting these issues.
“At the present time, SCImago is the only database that offers scientific productivity outcomes grouped by institutions. The use of other sources would therefore be much more cumbersome and problematic. This represents a limitation, since in fact SCImago is not exempt from criticism[37, 38]; it has been written that SCimago "omits a large amount of information, putting into question its transparency, reliability, and suitability for evaluative purposes in its current form, although most of the identified problems can be solved and might be the object of future improvements " [39]”
Obvious problems that come without comment here are
- the definition of Europe; it makes no sense that there is not a single university from the UK in the analysis [it may have left the EU but was there when the initial rankings were undertaken];
Thank you for the remark which is very appropriate. Taking advantage of the reviewer's suggestions, we have brought this consideration back within limits. We believe that in this way the text will be more understandable for the reader:
We also need to underline the way in which we have ranked European universities in relation to Brexit (and the UK universities). The survey was conducted with the aim of understanding the trend in scientific productivity of tacademe of the three major nation or federation of the world (for economic and scientific power) and of private companies. At the present time and for future projections of the European Union, this federation will have to do without UK universities (from 1 February 2020, the date of Brexit). Of course, if we consider the universities of the United Kingdom in 2010 and 2015, the position of the European Union would have been higher than without the UK (we have considered without the UK) and therefore, the decrease would have been even greater. In fact, a clear decrease also emerges in our analysis that did not consider UK universities even in the past. Our study therefore underlines that the tendency to lose positions in the European Union's world productivity ranking is also independent of Brexit. Furthermore, a multivariate analysis (taking into account the trend with and without the UK) would have been very problematic having to work on ranks and not on absolute values. Future studies would be very helpful in clarifying to what extent and whether Brexit may have weakened research in the European Union (and he United Kingdom)”.
However, we can anticipate to the reviewer that the UK trend is similar to that of the European Union if analyzed individually. That is, universities lost many places from 2015 to 2020. However, we repeat that the purpose of this work is to analyze the trend of the three major economic and scientific powers in the world and the UK no longer belongs to any of these three units.
- the inclusion of medical corporations such as Mayo in the university column, when it should be in the corporate column;
We have reviewed all the universities cited by SCImago and have decided to leave Mayo among the universities because, although it also has a health assistance network as its purpose, in its definition it is a "nonprofit American academic medical center" [see https: //www.mayoclinic. org / about-mayo-clinic / mission-values] and it is home to the top-ten ranked Mayo Clinic Alix School of Medicine https://college.mayo.edu/academics/mayo-clinic-alix-school-of- medicines /
However, we agree with the reviewer that all ranking systems obviously have limitations and all decisions like these can be refuted. We have better emphasized these aspects in the discussion and limits.
- the assumption that the hundreds of thousands of publications with multiple authors can be neatly attributed to hundreds of institutions. These are not the failings of the authors but they do indicate the limits to this taxonomy.
The reviewer is right, these are the limits we have highlighted better:
“Besides the limitations of SCimago, there are also the intrinsic limits to these bibliometric methodologies, which lie in the assumption that the hundreds of thousands of publications with multiple authors can be neatly attributed to hundreds of institutions and that the role of self-citations can somehow distort the results. These are not the failings of the present study but the limits of this approach with the current tools”.
All in all the results seem uninteresting and predictable as they stay at the level of the 4 groups [China, EU, US, Corporate].
Perhaps the results were predictable, but until very recently (we have cited contributions published in very important journals such as Fleming et al 2020 in Science and Poege et al 2019 in Sci Adv) it was thought that private companies were "not directly interested in investing in research, but rather appeared to take advantage of university research work ". This trend, which probably remains confirmed in “traditional" areas such as biomedical research, appears to be totally outdated in the world of e-research. This paper is the first to say it, so we therefore think that it may have an intrinsic interest.
To introduce some novelty, the authors could get into the weeds and for instance compare and contrast a single organization from each column and assess their contributions in terms of research and social impact.
The reviewer is right: some "case reports" may be interesting but precisely in function of the general trend, that is, which universities are gearing up to produce patents directly or in co-ownership? Which e-technology brands are gearing up to provide tertiary education and, if so, which in cooperation with universities?
However, we believe that these kinds of comparisons are interesting if you establish a general framework that just this very paper is offering.
We therefore believe that it is best to conduct case reports in future studies. Moreover, this paper would risk being too long (also owing to the limits imposed by the journal).
We also think the message needs to be clear because we believe it is important.
For example, as a case report at the top, there are some American universities such as Harvard or MIT that are apparently well contrasting the trend of losing positions and the growing importance of the innovation sector by creating joint ventures with private companies; while on the other hand there are mainly European universities located in poor geographical areas in terms of scarce presence of private companies, which have absolutely not adapted to the change and which, having reduced the possibility of accessing public funding, present an even more accentuated trend towards loss. However, since these observations to be dealt with more deeply would need methodologies of analysis different from those of the present study, we will only briefly mention these observations in the limits, referring to the need for further research and the development of specific methodologies for further study.
We have added in the limits section:
“The purpose of this work was to draw an overall picture and for this reason we could not go into detail in the analysis of those institutions that appear to be in contrast with the general trend, as well as those in which the trend appears to be very accentuated. The role of joint ventures with private companies in the first cases mentioned and the greater impact of the decrease in public funding in geographic poor areas in the second, deserve to be analyzed first with specific case reports and then with research that adopts a more specifically prepared methodology in the future”.
Round 2
Reviewer 3 Report
Thank you for engaging seriously with my concerns. I shall be interested to see how the published paper fares in the marketplace of ideas.